# Defining and identifying critical elements of, and lessons learned from addressing, 'operational readiness' for public health emergency events, including COVID-19: a scoping review protocol

Rene English ,[1] Juliet Charity Yauka Nyasulu ,[1,2] Karina Berner,[3] Heike Geduld,[4] Michael McCaul,[5] Conran Joseph,[3] Michele Pappin,[1] Nina Gobat,[6] Linda Lucy Boulanger,[6] Quinette Louw[3]

For numbered affiliations see end of article.

**Correspondence to**
Dr Juliet Charity Yauka Nyasulu; jnyasulu@sun.ac.za

## ABSTRACT

**Introduction** Much is known around public health preparedness and response phases. However, between the two phases is operational readiness that comprises the immediate actions needed to respond to a developing risk or hazard. Currently, emergency readiness is embedded in multiple frameworks and policy documents related to the health emergency cycle. However, knowledge about operational readiness' critical readiness components and actions required by countries to respond to public health eminent threat is not well known. Therefore, we aim to define and identify the critical elements of 'operational readiness' for public health emergencies, including COVID-19, and identify lessons learnt from addressing it, to inform the WHO Operational Readiness Framework.

**Methods and analysis** This is a scoping review following the Joanna Briggs Institute guidance. Reporting will be according to the Preferred Reporting Items for Systematic Reviews and Meta-Analyses Extension for Scoping Reviews (PRISMA-ScR) checklist. MEDLINE, Embase and Web of Science databases and grey literature will be searched and exported into an online systematic review software (eg, Rayyan in this case) for review. The review team, which apart from scoping review methodological experts include content experts in health systems and public health and emergency medicine, prepared an a priori study protocol in consultation with WHO representatives. ATLAS.ti V.9 will be used to conduct thematic data analysis as well as store, organise and retrieve data. Data analysis and presentation will be carried out by five reviewers.

**Ethics and dissemination** This review will reveal new insights, knowledge and lessons learnt that will translate into an operational framework for readiness actions. In consultation with WHO, findings will be disseminated as appropriate (eg, through professional bodies, conferences and research papers). No ethics approvals are required as no humans will be involved in data collection.

## STRENGTHS AND LIMITATIONS OF THIS STUDY

⇒ The COVID-19 pandemic has shown that, globally, countries even with well-resourced health systems and structured emergency preparedness plans in place were not able to sufficiently respond to the threat. Meaning that gaps existed between transition from preparedness to responding which is readiness. Therefore, defining and identifying critical elements of operational readiness for public health emergency events, including COVID-19, is critical.

⇒ Currently emergency operational readiness is embedded between preparedness and response and in most cases poorly defined. Therefore, we believe that an understanding of health systems readiness in responding to emergency is key.

⇒ The review team included members that provided a mix of methodological and content expertise that will aid decisions regarding a speed–rigour trade-off.

⇒ Currently, there is no clear definition of activities that constitute health systems emergency readiness and people use different names, while others name it either preparedness or response. In case these operational readiness definition words are not captured in the scoping review search strategy, this will be a limitation.

⇒ Limiting the search to English full texts and last 11 years may lead to and publication timeline biases.

**Protocol registration** This rapid scoping review has been registered on Open Science Framework (doi:10.17605/OSF.IO/6SYAH).

## INTRODUCTION

Much has been documented about how countries should best prepare to respond to health emergencies.[1–3] The effectiveness

of 'readiness'—a concept referring to actions needed to rapidly respond to an imminently anticipated risk or hazard—largely depends on the sufficiency and comprehensiveness of prior longer term 'preparedness' policies.[4] However, little is known about the critical components of readiness and the kinds of readiness actions that should be taken by countries at all levels in response to health emergencies. Such knowledge is critical to inform operational readiness actions for future events.

Health Emergencies and Disaster Risk Management (Health-EDRM) encompasses the intersecting fields of emergency and disaster medicine, health systems strengthening and resilience, disaster risk reduction, humanitarian response and community health resilience. Within this framework, it is accepted that the management of emergencies is a whole-of-society approach, focusing on all hazards and involving multiple sectors and multiple disciplines.[5] Health-EDRM involves four broad components, namely: (1) hazard vulnerability assessment (HVA) and mitigation; (2) preparedness; (3) response; and (4) recovery. Within these, the activities of 'readiness' will occur within both HVA and mitigation and preparedness components. These readiness activities are linked both temporally and structurally to a specifically identified hazard, whether that is an infectious disease or climate change event. Thus, what constitutes 'readiness' is determined by the nature of the hazard.

The WHO Strategic Framework for Emergency Preparedness[6] is a unifying framework for country-level public health emergency preparedness. This framework describes operational readiness to respond to emergencies as a continuous, coordinated process, involving a multisectoral response, incorporating multiple level infrastructure and following an all-hazard approach with a focus on high priority risks.[6]

The current COVID-19 global pandemic has exposed the fragility of health systems to respond to shocks in the form of disease outbreaks or health emergencies.[7] According to the WHO, the response of a public health system to an outbreak or health emergency such as the COVID-19 pandemic can be defined as a cycle that sways between preparedness and the actual response. Through applying a governance lens, the WHO has developed an Emergency Response Framework,[4] which describes the stages of an outbreak or health emergency. As alluded to previously, readiness to respond lies somewhere between preparedness and response; it is the instant action to an emergent or prominent risk and is hugely reliant on adequate preparedness.[4] In many instances, implementation of these well-designed disaster preparedness policies is met with significant challenges due to flaws in the 'readiness' of systems to do so. 'Readiness' as a concept has not been fully designed, and therefore, it is critical to define the critical components of readiness and the types of readiness actions to be taken in response to outbreaks and health emergencies to inform operational readiness actions for future events.[8] A preliminary search of MEDLINE, the Cochrane Database of Systematic Reviews,

Prospero and Joanna Briggs Institute (JBI) Evidence Synthesis revealed no current or underway systematic reviews or scoping reviews on the topic. The WHO is currently developing an Operational Readiness Framework intended to guide effective action. Specifically, the purpose of the framework is to scale-up preparedness for a specific risk at the local and national levels by considering how ready a country is to respond to the imminent threat and to identify key actions needed to be ready to respond effectively to that threat. To this end, WHO has called for a rapid scoping review to be conducted that will assist with defining available evidence related to readiness and readiness actions.

### Aim and objectives
The overarching aim of this rapid scoping review is to define and identify the critical elements of 'operational readiness' for public health emergencies, including COVID-19, and identify lessons learnt from addressing it, to inform the WHO Operational Readiness Framework.

To this end, the following objectives will be addressed:
1. To conceptualise and define 'operational readiness'.
2. To map and describe frameworks, policies and evidence/information related to 'operational readiness' for all hazards, with a strong focus on infectious diseases.
3. To define critical elements of 'operational readiness' at multiple levels of the health system (community, local, subnational, national, regional and global).
4. To identify lessons learned from enhancing or influencing 'operational readiness' (at multiple levels).

### Review question
#### Primary scoping review method question
The primary review question was formulated using the population, concept and context method[9]: *how can/do communities/countries/regions/global institutions operationalise readiness for imminent public emergencies?*

#### Subquestions
The review will seek to answer the following additional or subquestions:
1. How is 'operational readiness' for public health emergencies conceptualised and defined?
2. What are the critical elements (dimensions, operational actions and coordination) of 'operational readiness' for public health emergencies at multiple levels (community, local, subnational, national, regional and global)?
3. How did countries ready/prepare for COVID-19?
4. What lessons have been learned about 'operational readiness' during, for example, COVID-19/Ebola, with a strong focus on infectious disease emergencies?

### Eligibility criteria
#### Inclusion criteria
##### Participants/population
These are the groups or organisations who would respond and/or lead the response and include the following:

► Communities (local, subregional or national level).
► National, country, regional and global governments.
► Global health organisations, such as the WHO.

## Concept

The purpose of the scoping review is to define 'operational readiness'. This concept refers to the immediate action(s) that are taken to pre-position response actions needed to address a proximal, imminent hazard/threat, such as an 'acute' infectious disease outbreak or natural disaster threat (an all-hazards approach). These include but not limited to disease outbreaks, epidemics/pandemics, public health emergency, communicable diseases, incident management system, country risk profile and many other details. The concept lies between 'preparedness' and 'response'. To find evidence of readiness interventions, we will look at sources referencing preparedness, planning and disaster management as the term 'readiness' may be embedded in 'preparedness', or the term 'preparedness' may be used to describe actions that (based on our definition) we would describe as readiness.

We will consider sources that:
► Conceptualise, theorise, define or describe or interpret 'operational readiness' and/or preparedness for public health emergencies (at community, country regional or global levels) at the time when the threat of an infectious disease outbreaks or natural disaster becomes known, within a specific timeframe (namely, defining 'imminence').
► Contain explanations, descriptions, intervention approaches, analysis or frameworks or anticipatory actions for 'operational readiness' or preparedness for public health emergencies (at community, country, regional or global levels) when the threat of an infectious disease outbreaks or natural disaster becomes known.
► Provide the nature and description of critical elements (dimensions, coordination, roles of key stakeholders such as the community, health actors, policy makers, etc) of 'operational readiness' for public health emergencies at community, national, regional and global levels.

## Context

The context of health emergencies refers to natural disasters and infectious disease threats (new and re-emerging), that is, all hazards. Important to note is that these threats are acute (imminent) and impact the health of populations. These health emergencies occur within the community as well as health system and health service contexts.

The proposed definition of a 'health emergency' is an extraordinary event that is determined to 'constitute a public health risk whose scale, timing, or unpredictability threatens to overwhelm routine capabilities of the health system' (10 pS9) and potentially require a coordinated response at multiple levels.[10 11]

### Types of sources

► Peer-reviewed review or empirical research (any study design) that is available in full text and published in scientific journals between 2010 and 2021.
► Publicly available policy frameworks and programme reports.
  Published conference reports or electronic theses.
► Documents of which the full text or abstract is available in the English language. If the English version of the abstract is potentially eligible for inclusion, the full text (if German/French/Afrikaans) will be translated to make a final decision on eligibility.

## Exclusion criteria

► Papers focusing exclusively on longer term preparedness actions or exclusively on response actions will be excluded.
► Papers reporting on contexts beyond health emergencies or not focused on disease prevention and control will be excluded.

## METHODS AND DATA ANALYSIS

This rapid scoping review will be conducted in accordance with the JBI methodology for scoping reviews.[9] The review will be reported using the Preferred Reporting Items for Systematic Reviews and Meta-Analyses Extension for Scoping Reviews (PRISMA-ScR)[12] and PRISMA-S Extension for Searches in Systematic Reviews.[13]

## Search strategy

The search strategy will aim to locate peer-reviewed review or empirical research (any study design) that is available in full text and published in scientific journals, publicly available policy frameworks, programme reports and published conference reports or electronic theses. This will include humanitarian literature where health impacts or effects are the focus. Due to the rapid nature of the scoping review, we will restrict the search to studies published between 2010 and 2021 and those available in English (potentially eligible Afrikaans, German or French full texts, according to the English abstract, will be translated into English).

The electronic databases to be searched include MEDLINE, Embase and Web of Science. An initial limited search of MEDLINE was undertaken to identify articles on the review topic. The text words contained in the titles and abstracts of relevant articles, and the index terms used to describe the articles were used to draft a full search strategy for MEDLINE. The search strings and terms were developed iteratively and in consultation with the WHO and are centred around three key concepts: (1) emergencies/diseases/natural disasters; (2) readiness/preparedness/risk/planning; and (3) health systems/community. The search strategy, including all identified keywords and index terms, was subsequently adapted for Embase and Web of Science. Searches will be conducted by an expert information specialist in consultation with the review team. The reference list of all included sources

of evidence will also be screened for additional studies. Reporting of the searching will be guided by the PRISMA-S Extension for Searches in Systematic Reviews.[13]

## Searching other resources

Sources of unpublished studies/grey literature to be searched include various targeted repositories, websites and databases. These include global organisations (eg, the WHO, UNICEF, United Nations Office for Disaster Risk Reduction, United Nations International Strategy for Disaster Reduction, International Federation of Red Cross, International Committee of the Red Cross), regional WHO offices (ie, Southeast Asian, African, Western Pacific, Pan American, European and Eastern Mediterranean) and the European Centre for Disaster Medicine. Societies and organisations include the World Association for Disaster and Emergency Medicine, Médecins Sans Frontières and ReliefWeb. National websites include the US Centres for Disease Control and Prevention and Federal Emergency Management Agency, the Robert Koch Institute and Public Health England. Lastly, Evidence Aid will be included as an evidence repository.

## Selection of studies

All search hits will be imported into Rayyan V.0.1.0 software (Rayyan Systems Inc, Massachusetts, USA)[14] for screening, checking of duplicates and selection of final documents to be included. To support consistent abstract and title screening and refine eligibility, senior authors (RE, HG and MM) together with the title and abstract screeners (MP and Maria Yvone Charunbira), will (as an initial step) independently and in duplicate screen 100 articles, followed by discussion. The following proposed screening approach is adapted from the Cochrane Rapid Reviews Methods Group guidance for systematic reviews to balance rigour and speed consistent with rapid reviews.[15 16] Twenty per cent of titles and abstracts will be screened by two reviewers (MP and MYC), independently, in duplicate and with conflict resolution, to remove obviously irrelevant reports. After this, one reviewer (MP) will screen the remaining titles and abstracts, while the second reviewer (MYC) will verify excluded titles and abstracts and resolve conflicts.[15] If required, a third senior reviewer (HG or RE) will resolve any disagreements. The full texts of selected citations will subsequently be assessed in detail against the eligibility criteria by the first reviewer, while the second reviewer will verify all excluded full texts.[15] Reasons for exclusion of sources of evidence at full text that do not meet the inclusion criteria will be recorded. This information will be reported and added to a table of excluded studies in the scoping review. Any disagreements that arise between the reviewers at each stage of the selection process will be resolved through discussion, including with an additional senior reviewer (HG or RE) if needed. If study eligibility is unclear owing to missing data, further information will be requested from study authors. The results of the search and the source inclusion

process will be reported in full in the final scoping review and presented in a PRISMA-ScR flow diagram.[12]

## Data extraction and management

Due to the rapid design and potential large pool of included studies, we will use a dynamic approach to data extraction and management. For an included study yield of ≤25, data extraction will be done by one reviewer (MP), while a second reviewer (KB or MYC) will check for completeness and accuracy.[15] For yields between >25 but ≤75, two or more extractors will be used (eg, MP/KB/MYC/CJ/QAL/RE), while an additional reviewer will check for correctness and accuracy.[17] In the case of more than 75 included sources, we will consider a prioritisation process whereby we rank or stratify studies based on design and relevancy to the scoping review. Prioritised studies will then be included for data extraction until the review team, together with WHO, agrees that data saturation has been achieved. The reviewers will discuss the nature of the information that will be extracted before commencing the process to facilitate coherence. Any uncertainties before and during the extraction process will be discussed with team members to make a final decision.

The data extracted will include author name(s), publication year, publication country and World Bank classification, source classification as primary/secondary/multimethod, publication type, study design, aim/purpose, sample/facility description, method/tool for data collection, modifications to the data tool (if any), level (community, national, etc), type of emergency, operational readiness definition, preparedness definition, key actors, challenges/recommendations, lessons learnt and other relevant information/conclusions. In addition, data regarding readiness will be extracted according to the WHO's operational readiness components; these include:

► Leadership, governance and coordination.
► Country risk profile.
► Operational planning and coordination.
► Contingency finance.
► Health facility capacity and service delivery.
► Health workforce/human resources.
► Early warning or surveillance and health information systems.
► Community resilience and risk communications.
► Logistics or supply chain for access to essential medicines.
► WHO readiness.
► Partner readiness.

Framework details and any associated actions will be recorded. Finally, information regarding relevant models will be extracted, including URL links to figures/diagrams.

A draft extraction form will be pilot-tested independently by two reviewers using a sample of two to three potential included full-text articles/evidence sources.[17] Based on feedback from the two reviewers, the form may

be modified and revised as necessary during the process of extracting data from each included evidence source.[17] Necessary modifications will first be discussed within the review team for consensus, and any changes implemented will be reported in the final scoping review. Authors will be contacted where possible to clarify or obtain additional information.

## Methodological appraisal

Included peer-reviewed literature will be evaluated for quality based on appropriate pre-existing methodological quality checklists.

## Data analysis and presentation

Data will be synthesised in line with the core objectives of the rapid scoping review.

The included documents will be analysed using qualitative thematic analysis through an deductive synthesis approach.[18–20] We are proposing to use ATLAS.ti V.8 (Scientific Software Development GmbH) (https://atlasti.com/) to conduct thematic data analysis as well as store, organise and retrieve data. Data analysis will be carried out by the project group researchers, who have vast knowledge and experience in undertaking reviews, including scoping reviews, that have used qualitative thematic analysis.

Findings will be deductively coded into a conceptual model that is taken from the WHO Country Readiness for Health Checklist to define and identify the critical elements of 'operational readiness' for public health emergencies, including COVID-19, and identify lessons learnt from addressing it. We will also identify if there are additional consistent themes emerging from the analysis that are not currently included in the WHO Checklist, as potential additional items.

The analysis will start by evaluating documented text line by line, allocating text a descriptive label and code. The same will be done for the other focused questions on understanding the similarities and differences between operational readiness and preparedness and identifying critical elements. The researchers will remain close to the data from the primary sources when defining and understanding the meaning structure of these concepts and phenomena. Since the conceptual understanding of 'operational readiness' and 'preparedness' will be initially explored, described and theorised and may vary across sources, we will initially use broad, higher order codes (which may form main themes) developed deductively from the framework to organise the data. Once all data have been initially coded and collated, all the potentially relevant coded data extracts will be sorted and collated into themes and subthemes (including a 'miscellaneous' theme for codes that do not clearly fit into existing themes.[20] Senior reviewers (RE, HG and QAL) will debrief the researchers primarily responsible for the thematic analysis, and the review team will meet regularly to discuss codes and themes, including potential merging or further breakdown of themes (depending on whether there are enough data to support a theme, or the data are deemed too diverse). The themes will represent the synthesis and interpretation that go beyond the primary sources as well as deliver new insights and knowledge, which will translate into an operational framework for readiness and important lessons learnt.

A numerical description of the extent and nature of included evidence sources will be presented using tables and charts, accompanied by narrative summaries to describe how the results relate to the review's objectives.

## Patient public involvement

Patients or the public were not involved in the design, or conduct, or reporting, or dissemination plans of our research.

## Ethics, reporting and dissemination

No ethical approval is needed for this rapid scoping review, given that included sources will comprise of published and publicly available information.

The study was expected to commence in December 2021 to July 2022 with first scientific publication output expected in August 2022. The Stellenbosch University (SU) review team will work with the WHO commissioning group and draw on the expertise of expert advisors to the review team to produce the following outputs. Weekly internal and SU-WHO meetings have been conducted to provide input into the development of this research protocol and will continue to aid understanding of emerging insights and findings that can inform work tasks relevant to the technical product development. Interim findings from the rapid scoping review will be presented to the WHO. Following feedback, an updated interim report incorporating feedback from the WHO and expert advisory team will be presented. The final report of the full rapid scoping review will be delivered, along with a PowerPoint presentation to the WHO commissioning group of findings with talking points. In consultation with the WHO, findings will be disseminated further as appropriate (eg, through professional bodies, conferences and research papers). By defining evidence related to critical readiness components and actions, this review will reveal new insights, knowledge and lessons learnt that will translate into an operational framework for readiness actions.

## Author affiliations

[1]Division of Health Systems and Public Health, Department of Global Health, Faculty of Medicine and Health Sciences, Stellenbosch University, Cape Town, South Africa
[2]Department of Health Systems Strengthening, AFRIQUIP, Johannesburg, South Africa
[3]Division of Physiotherapy, Department of Health and Rehabilitation Sciences, Faculty of Medicine and Health Sciences, Stellenbosch University, Cape Town, South Africa
[4]Division of Emergency Medicine, Department of Family and Emergency Medicine, Faculty of Medicine and Health Sciences, Stellenbosch University, Cape Town, South Africa
[5]Centre for Evidence-based Health Care, Division of Epidemiology and Biostatistics, Department of Global Health, Stellenbosch University, Cape Town, South Africa
[6]Country Readiness Strengthening, World Health Organization (WHO), Geneve, Switzerland

**Acknowledgements** The rapid scoping review was commissioned by the WHO to inform an Operational Readiness Framework for the Country Readiness Strengthening Department in the World Health Emergencies Program in WHO (reference #: 2021/1145765; unit: MST; cluster: QNF/SCI).

**Contributors** RE is the principal investigator, and JCN coordinates the team, the research process and is a corresponding author. NG and LLB contributed to content through the process and provided guidance on the expected outcomes and WHO guidelines to be followed. KB and MP drafted the original protocol; MM, QAL, CJ and HG provided technical guidance on the methodology. All authors were involved in the conceptualisation of the concept paper and proposal writing and read the final version of the submitted protocol.

**Funding** This work is supported by the WHO through a consultancy fee and not research grant (reference: APW/RR/Readiness/2021/1145765); as a result, there is no grant number.

**Competing interests** None declared.

**Patient and public involvement** Patients and/or the public were not involved in the design, or conduct, or reporting, or dissemination plans of this research.

**Patient consent for publication** Not applicable.

**Ethics approval** Not applicable.

**Provenance and peer review** Not commissioned; externally peer reviewed.

**ORCID iDs**
Rene English http://orcid.org/0000-0002-1563-2532
Juliet Charity Yauka Nyasulu http://orcid.org/0000-0003-1158-6302

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
