## [Reviewer comments · BMJ Open]

ARTICLE DETAILS

TITLE (PROVISIONAL)	Defining and identifying critical elements of, and lessons learned from addressing, 'operational readiness' for public health emergency events, including COVID-19: A rapid scoping review protocol
AUTHORS	English, Rene; Nyasulu, Juliet; Berner, Karina; Geduld, Heike; McCaul, Michael; Joseph, Conran; Pappin, Michele; Gobat, Nina; Boulanger, Linda; Louw, QA

VERSION 1 – REVIEW

REVIEWER	Phattharapornjaroen, Phatthranit Sahlgrenska Academy, surgery
REVIEW RETURNED	06-May-2022

GENERAL COMMENTS	I am really thankful to be acknowledged for the existence of this work, really looking forward to reading the full manuscript. The topic was clearly defined and really interesting. The method was clearly described and rapid reviews are appropriate for the topic since it is a highly dynamic issue. Moreover, the number of reviewers might need to be more than 2 people. I am concerned that the search results may be in large amounts which might not be managed by 2 reviewers. Furthermore, I think the keywords for reviewing questions should be added, for example, preparedness, etc.
--

REVIEWER	Nelson, Christopher RAND Corp
REVIEW RETURNED	27-May-2022

GENERAL COMMENTS	The project and protocols seem well-conceptualized and appropriate to the task. The multilevel focus of the protocol is especially apt, given the complex intergovernmental dynamics at play in most major responses. I have just two small suggestions: (1) The description of the scope of disasters provided on p. 7 (i.e., "natural disasters and infectious disease threats") seems narrower than the list of search terms in the appendix would suggest; I suggest adjusting the text on p. 7 to reflect the breadth of "all hazards" evident in the appendix. (2) I don't recall seeing any clear indication the authors plan to look at defense/military sources. While the mission and operational context is often quite different, many military organizations have deep experience measuring, maintaining, and sustaining readiness and may provide useful insights for this work. This work may, indeed, come up via the authors' search terms, but I would encourage them to give this issue a little additional thought.
--

VERSION 1 – AUTHOR RESPONSE

Editor and reviewers' comments	Response
Please reformat the abstract so that it follows the structured abstract recommended in the journal's instructions for authors for research articles	Abstract revised, summarised to <300 words and recommended sections included
Please revise the 'Strengths and limitations' section of your manuscript (after the abstract).	This has been added between abstract and introduction
Please reformat the main text so that it follows the structure recommended in the journal's instructions for authors for study protocols, for example the main text of your manuscript should contain an Ethics and Dissemination section	Main text reformatted and Ethics and Dissemination sections added
Please include the planned start and end dates for the study in the methods section, as well as current study status.	Added in the methods section as below The study is expected to commence in December 2021 to July 2022 with first scientific publication out in August 2022.
A number of the tables in the supplementary information are formatted incorrectly when uploaded onto our system. We recommend you adjust these tables so they can easily be read.	Tables reformatted to figures
Kindly check the file format of the main document. The manuscript must be editable and submitted in Word Format	Done, work document submitted
The description of the scope of disasters provided on p. 7 (i.e., "natural disasters and infectious disease threats") seems narrower than the list of search terms in the appendix would suggest; I suggest adjusting the text on p. 7 to reflect the breadth of "all hazards" evident in the appendix	This statement has been added These include but not limited to Disease Outbreaks, epidemics/ or pandemics, public health emergency, communicable diseases, Incident Management System, country risk profile and many other as detailed in appendix 1 .
I don't recall seeing any clear indication the authors plan to look at defence/military sources. While the mission and operational context is often quite different, many military organizations have deep experience measuring, maintaining, and sustaining readiness and may provide useful insights for this work. This work may, indeed, come up via the authors' search terms, but I would encourage them to give this issue a little additional thought.	The protocol is in two phases, since this search has already been executed, we will include the defence military sources in the next phase.
Quality checks comments round 1	Response
Aside from the clean copy, please also provide a marked copy of your manuscript with 'tracked changes' and upload it under the file designation 'Main Document - marked copy'. This is to show all the changes you have made for your paper.	Done
Appendix I has been cited in the main text but not uploaded. Appendix II has been cited in the main text but not on the reference list.	This was a mistake it has been removed
Your submission should include a title page (embedded in the main document) which must contain the following information after the title	Done
The name, postal address, e-mail, telephone, and fax numbers of the corresponding author.	Done
The full names, institutions, city, and country of all co-authors.	Done

Up to five keywords or phrases suitable for use in an index (it is recommended to use MeSH terms).	Done
Word count - excluding title page, references, figures and tables.	Done
Please write a correct format of abstract for protocol paper. Below is your guide	Done
Please provide an 'Article summary' section consisting of the heading: 'Strengths and limitations of this study'. Please note that 'Strengths and limitations of this study' should consist of 3-5 bullet points placed after abstract section	Done
Please provide a detailed contributorship statement	Done
Quality checks round 2	
Author order in the ScholarOne system is different from the main document. Kindly ensure that the arrangements of authors in your main document and ScholarOne submission system are the same.	This has been sorted
Please write a correct format of abstract for protocol paper and ensure that the abstract in ScholarOne is also updated. Below is your guide: Introduction, Methods and Analysis and Ethics and Dissemination	Done see revised abstract
Please ensure to have the same contributorship statement both in main document and ScholarOne where all the names/initial of authors are mentioned in it.	Done
Please move the 'Patient and Public Involvement' statement under 'Methods' section.	Content moved to methodology section
Quality checks round 3	
Missing heading for 'Patient and Public Involvement' in the methods section.	Included
Please provide a more detailed contributorship statement. It needs to mention all the names/initials of authors along with their specific contribution/participation for the article. This should be stating how each author contributed to the article. It should discuss on the planning, conduct and reporting of the work in your paper. You may also consider the conception and design, acquisition of data or analysis and interpretation of data, etc. The statement in the ScholarOne system and main document should matched. Please see link below for more information: https://authors.bmj.com/policies/bmj-policy-on-authorship	Done
In word version, please provide a point-by-point response to the Editor's comments and reviewer's comments.	Done
The list of authors in the uploaded authorship form and in your main document file is a mismatched.	Revised and amended

VERSION 2 – REVIEW

REVIEWER	
REVIEW RETURNED	
GENERAL COMMENTS	

VERSION 2 – AUTHOR RESPONSE

VERSION 3 – REVIEW

REVIEWER	
REVIEW RETURNED	

GENERAL COMMENTS	
--

REVIEWER	
REVIEW RETURNED	

GENERAL COMMENTS	
--

VERSION 3 – AUTHOR RESPONSE

VERSION 4 – REVIEW

REVIEWER	
REVIEW RETURNED	

GENERAL COMMENTS	
--

REVIEWER	
REVIEW RETURNED	

GENERAL COMMENTS	
--

VERSION 4 – AUTHOR RESPONSE